# 1-Methylcyclopropene and UV-C Treatment Effect on Storage Quality and Antioxidant Activity of ‘Xiaobai’ Apricot Fruit

**DOI:** 10.3390/foods12061296

**Published:** 2023-03-18

**Authors:** Yunhao Lv, Anzhen Fu, Xinxin Song, Yufei Wang, Guogang Chen, Ying Jiang

**Affiliations:** 1College of Food Science, Shihezi University, Shihezi 832003, China; 2Research Center of Xinjiang Characteristic Fruit and Vegetable Storage and Processing Engineering, Ministry of Education, Shihezi 832000, China

**Keywords:** UV-C, 1-MCP, apricot, antioxidant activity, physicochemical characterization

## Abstract

The ‘Xiaobai’ apricot fruit is rich in nutrients and is harvested in summer, but the high temperature limits its storage period. To promote commercial quality and extend shelf life, we investigated the effectiveness of Ultraviolet C (UV-C) combined with 1-methylcyclopropene (1-MCP) treatment on ‘Xiaobai’ apricot fruit stored at 4 ± 0.5 °C for 35 days. The results revealed that the combination treatment of 1-MCP and UV-C performed better than either UV-C or 1-MCP alone in fruit quality preservation. The combination treatment could delay the increase in weight loss, ethylene production, and respiration rate; retain the level of soluble solid content, firmness, titratable acid, and ascorbic acid content; promote the total phenolics and flavonoids accumulation; improve antioxidant enzyme activity and relative gene expression, and DPPH scavenging ability; and reduce MDA, H_2_O_2_, O_2_^.−^ production. The combined treatment improved the quality of apricot fruit by delaying ripening and increasing antioxidant capacity. Therefore, combining UV-C and 1-MCP treatment may be an effective way to improve the post-harvest quality and extend the storage period of the ‘Xiaobai’ apricot fruit, which may provide insights into the preservation of ‘Xiaobai’ apricot fruit.

## 1. Introduction

The ‘Xiaobai’ apricot is a characteristic of agricultural products in Xinjiang, China. It has a high nutritional value and a pleasant taste and is enjoyed by the locals [1]. However, because of the mature season in summer and its origin in southern Xinjiang, the high ambient temperature makes long-term storage and long-distance transportation difficult [2]. Therefore, it is critical to determine how to extend the storage period of the ‘Xiaobai’ apricot while maintaining its commercial quality. So far, several apricot fruit preservation methods have been proposed, including coating preservation, salicylic acid treatment, calcium treatment, and near-freezing temperature storage [3,4,5,6].

Ultraviolet-C (UV-C) irradiation (200–280 nm), a non-thermal disinfection method used in fruits and vegetables, can cause DNA injury in microorganisms by altering pyrimidine dimer formation, which can prevent agricultural product decay [7]. Moreover, UV-C is an abiotic stress that could damage the biological membrane and stimulate the generation of reactive oxygen species (ROS). It can also stimulate the antioxidant system and promote secondary metabolite synthesis [8,9]. In fruits and vegetables, phenolic and flavonoid compounds are important secondary metabolites with antioxidant properties. It was previously reported that UV-C treatment improved the accumulation of phenolic and flavonoid compounds in strawberries, sweet cherries, and blueberries [10,11,12]. In addition to antioxidant compounds, excess ROS would stimulate the expression of enzymatic plant antioxidant systems such as catalase (CAT), superoxide dismutase (SOD), peroxidase (POD), and ascorbate peroxidase (APX) [13]. Rivera–Pastrana et al. [14] revealed that UV-C irradiation increased the activity of SOD, POD, and CAT in papaya fruit. In contrast, Sripong et al. [15] reported that UV-C irradiation promoted ROS production while also increasing the activity of POD in mangosteen. However, several studies demonstrated that UV-C irradiation increased the respiration rate and ethylene production in white asparagus, zucchini, and tomato [16,17,18]. Ethylene production would accelerate fruit ripening and senescence, which are detrimental to long-term fruit storage. The 1-methylcyclopropene (1-MCP) may inhibit ethylene response by binding specifically to ethylene receptors, delaying fruit and vegetable maturation and senescence [19,20]. Previous studies revealed that 1-MCP treatment could effectively inhibit the ethylene effect, delaying ripening and senescence in pears and pomegranates [21,22,23]. In addition, 1-MCP was found to be effective in retaining antioxidant enzyme activity and alleviating ROS damage in apples, bitter melon, and nectarine [24,25,26]. Therefore, the UV-C and 1-MCP treatments effectively improved the storage quality and extended duration of post-harvest fruit and vegetables.

To date, no studies have been conducted to investigate the effects of UV-C irradiation combined with 1-MCP treatment on apricot fruit storage. The present study aimed to explore the efficiency of UV-C irradiation combined with 1-MCP treatment on apricot post-harvest storage and preservation and to provide a theoretical basis for extending the storage period and elevating the quality of post-harvest apricot fruit.

## 2. Materials and Methods

### 2.1. Plant Material and Experimental Design

The ‘Xiaobai’ apricot with low maturity (soluble solid content (SSC: 11 ± 0.5%, firmness: 23.5 ± 0.5 N) was collected on June 2020 from a plantation in Bugur County, Xinjiang, China. On the day of harvest, the apricot fruit was transported to the laboratory of Shihezi University. After a 24 h precooling period, fruit with uniform size, no mechanical damage or diseases, and a similar maturity were selected for experiments. The selected apricot fruit was then randomly distributed into four groups (each group had 25 kg apricot fruit) as follows: (1) control fruit with no treatment; (2) 1-MCP treatment, the sample apricot fruit was fumigated in a 1 m^3^ airtight container for 20 h at 20 °C, with a 1-MCP concentration of 1 μL L^−1^; (3) UV-C treatment, the sample apricot fruit was placed on a clean bench (length: 1.3 m, width: 0.66 m, height: 0.52 m). The fruit was exposed to a UV-C lamp (30 W/G30T8, Philips) for 5 min before being rotated 180° for another 5 min to achieve a total irradiation dose of 1.25 kJ m^−2^. A UV radiation meter (LS126C, Linshangtech, China) was used to measure the UV dose. (4) Combined treatment, fruits were treated with 1-MCP as the method of 1-MCP treatment for 20 h, then treated with UV-C irradiation as the method of UV-C treatment. Following treatment, all four fruit groups were stored in polystyrene foam boxes at 4 ± 0.5 °C and 80 ± 5% relative humidity for 35 days. At each sampling time point (day 0, 7, 14, 21, 28, and 35) during storage, 4 kg fruit from each treatment group was taken for triplicate analysis.

### 2.2. Determination of Ethylene Production and Respiration Rate

Gas chromatography (GC-16A, Shimadzu, Kyoto, Japan) was used to measure the ethylene production of ‘Xiaobai’ apricot fruit. A total of 500 g of apricot fruit was placed in a 1 L closed chamber for 1 h, and an injection needle absorbed 0.1 mL of gas. The following were the GC conditions: The column and detector temperatures were 50 °C and 150 °C, respectively. The nitrogen (N_2_) flow rate was 18 mL min^−1^. The ethylene production rate is expressed in ng kg^−1^ s^−1^ as the amount of ethylene produced by 1 kg fruit per unit time.

A sample of 1 kg of fruit was randomly selected and placed in a 1 L sealed tank attached to a carbon dioxide tester (CES-10, Zhonggu, Shanghai, China) for 30 min at 4 °C. Respiration rate was expressed by CO_2_ production rate in ng kg^−1^ s^−1^.

### 2.3. Firmness, Weight Loss, Soluble Solid Content and Titratable Acid

Twenty ‘Xiaobai’ apricot fruits were randomly selected, peeled, and placed on the durometer platform. The durometer was vertically inserted into the scale line by pressing the operating lever uniformly. The firmness was expressed as the average reading value in N.

The weight loss was measured using the weighing method, with 100 fruits selected randomly in each treatment group and weighed every seven days. The differential weight method was used to calculate the weight loss. The calculation formula is as follows:$$\mathrm{weight}\;\mathrm{loss}\;(\%)=(\frac{\mathrm{initial}\;\mathrm{weight}-\mathrm{final}\;\mathrm{weight}\;}{\mathrm{initial}\;\mathrm{weight}})\times 100\%$$

Samples of 10 g of fruit were placed in gauze and squeezed to determine soluble solid content (SSC), and the juice was dropped into a digital refractometer (A1701161, ATAGO, Tokyo, Japan). Each group was repeated ten times, and the results were expressed as the average reading value in percentage (%).

Acid-base titration was used to determine titratable acid (TA). A total of 10 g of fruit was homogenized in a mortar with deionized water, transferred to a flask, and extracted for 30 min. Then, 10 mL supernate was filtrated into a triangular flask for acid-base titration with 2 mol L^−1^ NaOH solution. TA was calculated with the malic acid degree, and the result was expressed in percentage (%).

### 2.4. Ascorbic Acid, Total Phenolics, and Flavonoid Content

The detection of ascorbic acid (ASA) content was based on the work of Xu et al. [27]. The samples of 10 g of fruit were triturated, extracted with oxalic acid, and then titrated with 2,6-dichloroindophenol until the pink color appeared. The result was expressed in mg kg^−1^.

Total phenolics (TP) content and flavonoids were detected by the method described by Li et al. [28]. Briefly, 2 g fruit samples were ground and homogenized in 1% HCl-methanol solution in an ice bath, then extracted at 4 °C for 20 min. TP was measured using the optical density (OD) at 760 nm. Gallic acid was used as a standard curve to express the TP content, which was expressed as g kg^−1^. To measure total flavonoids, 1 mL of supernate was transferred to a test tube, and 1 mL 5% (*w*/*v*) NaNO_2_ and 0.25 mL 10% (*w*/*v*) AlCl_3_ were added to the tube, respectively. After 5 min, 1 mL 1 mol L^−1^ NaOH was added to the mixture. As flavonoids, the OD was measured at 510 nm. To express the flavonoid content, a standard curve was made with different concentrations of rutin.

### 2.5. Peroxidase, Ascorbate Peroxidase, Superoxide Dismutase and Catalase Activities

Peroxidase (POD) activity was measured by the method described in a study by Sheng et al. [29]. Five grams of apricots were homogenized in 5 mL of 50 mM pH 7.8 acetate buffer (containing 1 mM PEG, 4% PVPP, and 1% Triton X-100) and centrifuged at 8000× *g* for 20 min at 4 °C. The 0.5 mL supernatant was collected and added with 3.0 mL 25 mmol L^−1^ guaiacol and 200 μL 0.5 mol L^−1^ H_2_O_2_. The change of OD within 2 min was measured at 470 nm, and expressed with the unit of U kg^−1^.

Ascorbate peroxidase (APX) activity was detected by a method described by Xu et al. [27]. Five-gram samples were homogenized in 5 mL of 50 mM pH 7.5 potassium phosphate buffer (containing 0.1 mmol L^−1^ EDTA, 1 mmol L^−1^ ascorbic acid, and 2% PVPP) and centrifuged at 8000× *g* for 20 min at 4 °C. The 0.5 mL supernatant was collected and mixed with 0.3 mL 2 mmol L^−1^ H_2_O_2_. The change in OD within 2 min was measured at 290 nm.

Superoxide dismutase (SOD) activity was measured by the procedure explained by Xu et al. [30]. Five-gram samples were homogenized in 5 mL of 50 mM pH 7.8 sodium phosphate buffer (containing 5 mmol L^−1^ DTT and 5% PVP) and centrifuged at 8000× *g* for 20 min at 4 °C. The 0.5 mL supernatant was collected and mixed with 100 μM EDTA-Na_2_ (0.15 mL), 750 μM nitro-blue-tetrazolium (NBT) (0.15 mL), 130 mM methionine (0.15 mL), and 20 μM riboflavin (0.15 mL). The change in OD within 2 min was measured at 560 nm.

Catalase (CAT) activity detection was based on the method described by Li et al. [28]. The extraction method is identical to that of SOD. The enzymatic reaction system included 2.9 mL of 20 mmol L^−1^ H_2_O_2_ and 100 µL of enzyme extraction solution. The change in OD within 2 min was measured at 240 nm.

### 2.6. Malondialdehyde, Hydrogen Peroxide Content, and Superoxide Radicals Generation Rate

Malondialdehyde (MDA) content was measured through a method explained by Fan et al. [3]. Five-gram apricot samples were triturated in 15 mL of 5% (*w*/*v*) trichloroacetic acid (TCA) and centrifuged at 8000× *g* for 20 min at 4 °C. A total of 1 mL supernate was extracted and mixed with 3 mL 0.5% (*w*/*v*) thiobarbituric acid (TBA) containing 10% TCA and incubated in a 100 °C water bath for 20 min, and then centrifuged at 8000× *g* for 10 min at 4 ˚C. The OD of the supernate was measured at 450, 532, and 600 nm, respectively, and the following equation calculated MDA content:$$\mathrm{MDA}\;\mathrm{content}=\frac{\left[6.45\times \left({\mathrm{OD}}_{532}-{\mathrm{OD}}_{600}\right)-0.56\times {\mathrm{OD}}_{450}\right]\times V}{V_{s}\times m\times 1000}$$
where V is volume of the extracted sample, Vs is volume of the determination solution, and m is the mass of the sample. The unit of the result was mmol kg^−1^.

The hydrogen peroxide (H_2_O_2_) content was determined according to the method described by Xu et al. [30] with slight modification. Two-gram samples were homogenized with 5 mL acetone (100%) and centrifuged at 5000× *g* for 20 min at 4 °C. A 1 mL supernatant sample was extracted and mixed with 0.1 mL titanium tetrachloride-HCl and 0.2 mL ammonium hydroxide, followed by centrifugation at 8000× *g* for 10 min at 4 ˚C. The H_2_O_2_ content was expressed by measuring OD at 412 nm in mol kg^−1^.

The method of Li et al. [28] described was referenced to measure the superoxide radicals (O_2_^.−^) generation rate. Five-gram apricot samples were ground with 5 mL of 50 mM pH 7.8 phosphate buffer and centrifuged at 5000× *g* for 20 min at 4 °C. The supernate was incubated for 1 h at 25 °C with 1 mL 1 mM hydroxylamine hydrochloride, then mixed with 1 mL 17 mM 4-aminobenzene sulfonic acid and 1 mL 7 mM α-naphthylamine. The mixture was blended and incubated for 25 min at 25 °C for the chromogenic reaction. The OD was measured at 530 nm. The result was described as U kg^−1^.

### 2.7. 2,2-Diphenyl-1-Picrylhydrazyl Radical Scavenging Activity Assay

The detection of 2,2-diphenyl-1-picrylhydrazyl (DPPH) radical scavenging rate was based on the method described by Zhang et al. [31]. After homogenizing 1 g of apricot fruit with 80% methanol, the sample was centrifuged. Supernate (0.4 mL) was mixed with 2.4 mL DPPH solution in a tube and kept in the dark at 25 °C for 30 min. The OD was measured at 517 nm. The sample extraction was replaced with 80% methanol in the blank control. The DPPH radical scavenging rate was calculated by the following equation:$$\mathrm{DPPH}\;\mathrm{radical}\;\mathrm{scavenging}\;\mathrm{activity}=(1-\frac{A_{s}}{A_{b}})\times 100\%$$

A_s_ and A_b_ are the absorbance of the sample and blank, respectively.

### 2.8. Reverse Transcription–Quantitative PCR (RT-qPCR)

The RNA extraction kit (CW2598S, CWbio, Taizhou, China) extracted RNA from 0.1 g of frozen powdered apricot sample. The total RNA content was determined by UV-spectrophotometer (NANO 2000, Thermo, Waltham, MA, USA). Then, a cDNA synthesis kit (D7160L, Beyotime Biotechnology, Haimen, China) was used to synthesize cDNA from 2 μg of total RNA. Primer 5.0 (Premier Biosoft, San Francisco, CA, USA) was used to design the primer sequences for q-PCR and listed in Table 1. For normalization, the BoActin protein gene was used as a reference gene. RT-qPCR reactions were carried out in a fluorescent quantitation PCR amplifier (Exicycler 96, Bioneer, Daejeon, Republic of Korea), and the relative gene expression was determined using the 2^−ΔΔCt^ method.

### 2.9. Statistical Analysis

The statistical analysis of the obtained data was calculated by IBM SPSS Statistics 20 software. Duncan’s multiple range test was used to calculate the significant differences. The results of two samples were regarded as a significant difference when *p*-value less than 0.05. Results of each sample are expressed as the mean ± standard deviation (SD). All experimental data were plotted using Origin 2019.

## 3. Results

### 3.1. Respiration Rate and Ethylene Production

The respiration rate in the four groups of apricot fruit increased and then decreased during storage (Figure 1a). On day 21, the respiration rate in untreated and UV-C-treated fruit peaked, but the respiration rate in UV-C-treated fruit was higher than the control. During the storage period, the respiration rate in the 1-MCP treatment group was always lower than the control, with the peak occurring on day 28. Except for the first seven days of storage, the respiration rate in the combined treatment group was lower than the control but higher than the 1-MCP treatment group, and the peak was also delayed on day 28.

The ethylene production trend in post-harvest apricots followed the same pattern as the respiration rate (Figure 1b). On day 21, ethylene production in UV-C-treated fruit was higher than in untreated fruit. Furthermore, during storage, ethylene production in the combined treatment group was higher than in the 1-MCP treatment group but lower than in the UV-C and control groups.

### 3.2. Firmness and Weight Loss

The firmness of untreated apricot fruit decreased continuously, reaching a minimum (8.31 N) on day 35 (Figure 1c). Compared to untreated samples, applying UV-C or 1-MCP maintained the firmness of apricot fruit, though 1-MCP maintained greater firmness than UV-C-treated fruit. Furthermore, the firmness of the combined treatment (12.39 N) was higher than that of the single treatment and the control, which were 1.06, 1.18, and 1.49 times compared to 1-MCP, UV-C treated, and untreated samples, respectively.

The weight loss of untreated fruit increased steadily during storage, reaching 19.70% on day 35 (Figure 1d). The three treatments could restrain the weight loss in apricot at the end of storage, and the weight loss in 1-MCP, UV-C, and combined treatment were 15.17%, 17.64%, and 13.06%, respectively, which were lower than that in the control. Furthermore, in the first seven days, the weight loss in the UV-C and combined treatments was higher than in control.

### 3.3. SSC and TA

SSC in untreated apricot fruit increased from day 0 to 21, then decreased until the end of storage (Figure 1e). SSC in UV-C-treated fruit followed the same trend as in untreated samples but was higher. Before day 21, the SSC in the 1-MCP and combined treatments were lower than in the untreated and UV-C samples but increased on day 28 and was higher than in the UV-C-treated and untreated fruits.

The TA in the four groups of apricot fruit decreased during the storage (Figure 1f), but the TA in the control group decreased more quickly than the other three treatment groups. The decrease rate of TA in the UV-C treatment group was the fastest among the three treatment groups, followed by the 1-MCP treatment group, and the slowest in the combined treatment group. After the storage period, the TA in the UV-C, 1-MCP, and the combined treatment groups were 1.07, 1.22, and 1.34 times that of the control, respectively.

### 3.4. ASA, TP, and Flavonoid Content

The ASA content in the control and treatment groups decreased continuously with the increase in storage time (Figure 2a). The control group decreased rapidly, reaching 16.58 mg kg^−1^ on day 35. On day 35, the ASA content in 1-MPC- and UV-C-treated fruit was 21.02 and 23.41 mg kg^−1^, respectively, delaying the decrease in ASA content during the storage. Similarly, the combined treatment group had an ASA content of 26.07 mg kg^−1^ on day 35, which was higher than the two single treatment groups, and 1.57 times that of the control.

The TP content in untreated fruit increased with storage time, peaking on day 21 (1.79 g kg^−1^), and then decreased (Figure 2b). TP content in 1-MCP-treated fruit was higher than untreated fruit on days 21 and 35, implying that 1-MCP treatment could delay the TP content decrease during the late storage period. On days 7 to 21, the UV-C and the combined treatments could induce the accumulation of total phenolic content. The TP in the UV-C treatment and the combined treatment groups was 1.21 and 1.28 times that of the control group at day 21, respectively. Furthermore, the decrease in TP in combined treated fruit was slighter than in UV-C-treated fruit, and the highest content was retained at the end of storage. The flavonoid content change was similar to TP content in each fruit group (Figure 2c). On day 21, the flavonoid content in untreated fruit peaked at 19.86 mg kg^−1^, which was 4.47%, 11.10%, and 14.36% lower than 1-MCP, UV-C, and the combined treated fruit, respectively.

### 3.5. POD, CAT, SOD, and APX Activities and Corresponding Relative Gene Expression

Figure 3a depicts that the POD activity in the control group increased continuously, reached the maximum on day 14, and then decreased. POD activity in three treated samples followed a similar pattern, but the peak was delayed until day 21. Furthermore, POD enzyme activity in the UV-C treatment and combined treatment groups increased from day 7 to 21, reaching 1.18 and 1.23 times that of the control on day 21. The 1-MCP treatment group had no apparent effect on improving POD enzyme activity from day 0 to 7, but did improve POD activity from day 14 to 35. Figure 3b indicates that the relative expression of *pmPOD* was higher in the UV-C and combined treatment groups than in the control group during storage.

As shown in Figure 3c, CAT activity in the control group generally increased during the initial storage stage and reached the maximum on day 14, then gradually decreased with a temporary increase (day 28). CAT activity was higher in 1-MCP- and UV-C-treated fruit than in control during storage; moreover, CAT activity in UV-C treatment was higher than in 1-MCP treatment. On day 21, the CAT activity in the combined treatment group was 5.22%, 13.61%, and 27.52% higher than in the UV-C treatment, 1-MCP treatment, and control groups. Similar to the activity of CAT, the expression of CAT in the four groups increased for the first 21 days and then decreased (Figure 3d). However, the expression of *pmCAT* in UV-C treatment and combined treatment was higher than the control.

Figure 3e,g indicate the changes in SOD and APX enzyme activities. SOD and APX enzyme activities in untreated fruit increased continuously, reaching the maximum on day 28, and then decreased. UV-C or 1-MCP treatment, like POD and CAT, maintained a higher SOD and APX activity level in apricot fruit than in untreated fruit. Moreover, APX and SOD activity in the combined treatment group was 8.52 and 5.81 U g^−1^ on day 28, respectively, and improved by 25.2% and 35.6% compared to the untreated. The change of *pmSOD* (Figure 3f) and *pmAPX* (Figure 3h) expression during storage was more significant in the UV-C and combined treatments than in the control and 1-MCP treatments, which was similar to the activity of SOD and APX.

### 3.6. MDA, H_2_O_2_ Content, and O_2_^.−^ Generation Rate

During storage, the MDA content in four groups of apricot fruit increased continuously (Figure 4a). The MDA content in the untreated group reached its maximum on day 35 (1.84 nmol kg^−1^). The MDA content in the UV-C and combined treatment groups was higher than the control group for the first 14 days, then the rise slowed down from day 21 to 35, reaching 1.76 and 1.68 nmol kg^−1,^ respectively, on day 35, amounts which were lower than the control. During storage, MDA content in the 1-MCP treatment group increased slowly and was always lower than in other treatment groups, reaching 1.64 nmol kg^−1^ on day 35.

The H_2_O_2_ content in the four groups of apricot fruit increased with storage time (Figure 4b), with the untreated fruit reaching 96.97 mol kg^−1^ on day 35. The H_2_O_2_ content in the 1-MCP treatment group was 75.76 mol kg^−1^ on day 35, which was lower than the control group. However, H_2_O_2_ content in the UV-C treatment group was higher than the control on days 7 and 14, then gradually increased slowly to 89.51 mol kg-1 on day 35, which was 7.7% lower than the control. The combined treatment group had higher H_2_O_2_ content than the 1-MCP treatment group for the first seven days, then the increase slowed and was the lowest compared to the other three groups on day 35 (72.12 mol kg^−1^).

Figure 4c indicates that the O_2_^.−^ generation rate in untreated fruit increased with storage time, reached a maximum on day 21 (174.65 U kg^−1^), and then decreased to 157.29 U kg^−1^ on day 35. The O_2_^.−^ generation rate trend in the 1-MCP treatment group was similar to the control group but lower during storage (*p* < 0.05) than control. The UV-C treatment group had a higher O_2_^.−^ generation rate than the control on days 7 to 14, but decreased rapidly and was lower than in the control group from day 21 to 35. The combined treatment group demonstrated the same pattern as the UV-C group but was higher than the control group on day 7 and lower than the two single treatment groups from day 21 to 35.

### 3.7. DPPH Scavenging Rate

The DPPH scavenging rate in untreated fruit increased with storage time, peaked on day 21, and began to fall (Figure 4d). During storage, the DPPH scavenging rate in all three treatment groups followed the same pattern and was higher than the control. When the peak values in each group on day 21 were compared, the DPPH scavenging rate in the 1-MCP, UV-C, and combined treatment groups was 13%, 26.7%, and 30.1% higher than the control, respectively. The findings revealed that the three treatments could efficiently improve the DPPH scavenging rate of ‘Xiaobai’ apricot fruit during storage, with the UV-C treatment and combined treatment having a more effective increase in DPPH clearance rate.

### 3.8. Correlation Analysis

Figure 5 depicts the correlation analysis result between each detected index during storage. The results indicated that TA and firmness were negatively correlated with ethylene production. In contrast, weight loss, SSC, and MDA were positively associated with ethylene production. Weight loss and ROS (H_2_O_2_ and O_2_^.−^) levels were positively correlated with respiration rate. Moreover, antioxidant enzyme activity and secondary metabolites (TP and flavonoid) content were positively associated with ROS, whereas ASA content was negatively correlated with ROS. Furthermore, antioxidant ability (DPPH scavenging rate) was positively linked to antioxidant enzyme activity and secondary metabolite content.

## 4. Discussion

As a typical climacteric fruit, the ‘Xiaobai’ apricot fruit had a clear respiration rate and ethylene production peaks, two important nodes for the climacteric fruit [32]. Fruit and vegetables had the best edible quality in a short period following the appearance of respiration peak, whereas ethylene production can accelerate fruit ripening and senescence. Delaying the respiratory peak and reducing ethylene production is important for extending the fruit storage period [33]. The present study indicated that the respiration rate and ethylene release of UV-C-treated fruit increased sharply at the start of storage, which may be related to the physiological damage caused by UV-C irradiation promoting plant self-healing and energy supply–demand [34]. However, 1-MCP treatment effectively inhibited fruit respiration rate and ethylene production, which were linked to the specific combination of 1-MCP and ethylene receptors in fruit [21]. Concurrently, this effect is also reflected in the combined treatment. Xu and Liu [35] reported that blueberry fruit treated with 1-MCP and UV-C had lower levels of respiration and ethylene production. Tiecher et al. [9] indicated that ethylene release in tomato fruit treated with the combination of 1-MCP and UV-C treatment was lower than the UV-C treatment alone, which was consistent with our findings. Therefore, the combined treatment has the potential to effectively reduce the increase in respiration rate and ethylene release caused by UV-C irradiation.

SSC and TA are important for determining the edible and flavor quality of harvested fruit. Moreover, the ratio of SSC to TA is frequently used to indicate ripeness in fruit. Apricot ripening, like other fruits, is typically accompanied by the accumulation of sugar and the degradation of organic acid [36]. In addition, the firmness of the apricot fruit reflected its maturity. Fruit softening during post-harvest ripening would result from the degradation of cell wall components such as pectin and cellulose [37,38]. In the present study, it was found that UV-C irradiation accelerated the accumulation of SSC in fruit, which could be attributed to the promotion of fruit ripening by ethylene production [39]. During the post-ripening of apricot fruit, 1-MCP could effectively restrain the increase in SSC and inhibit the decomposition of TA. Previous studies have revealed that 1-MCP treatment can effectively prolong the post-harvest ripening process of the French prune while hindering the increase in SSC [40]. Similarly, the rise of SSC and decreased TA was slower in combined treated apricot fruit, while the TA was higher in single-treatment groups. Therefore, when compared to a single UV-C irradiation treatment, the combined treatment could more effectively delay the apricot fruit post-harvest ripening.

Phenols and flavonoids are important secondary metabolites that contribute significantly to improving the antioxidant capacity of fruit [41]. The present study indicated that the combined treatment could effectively limit the decrease in ASA content and accelerate the production of phenols and flavonoids. In two single treatments, 1-MCP suppressed the reduction in total phenolic and flavonoid contents but did not affect production. In contrast, UV-C promoted the accumulation of total phenolics and flavonoids, implying that UV-C irradiation can accelerate the secondary metabolic rate of fruits during storage [42,43]. Wang et al. [12] described that UV-C irradiation could accelerate the accumulation of phenolics and flavonoids in blueberries. Therefore, it is necessary to recognize that UV-C treatment was more important than 1-MCP treatment in promoting total phenolics and flavonoid content in combined treatment. ASA is an important nutrient in apricot fruit and an antioxidant-active substance [44]. The present study revealed that all three treatments could effectively delay the decrease in ASA content, with the combined treatment having the best effect, possibly due to the fact that the secondary metabolism and antioxidant capacity of fruit was enhanced by UV-C. Ávila–Sosa et al. [45] reported that pre-treatment of hawthorn with UV-C irradiation could maintain the ASA content during storage. In contrast, Xiong et al. [40] indicated that ASA content in prune would be restrained by being treated with 1-MCP. Therefore, combining 1-MCP with UV-C treatment could be an efficient way to limit ASA degradation and promote the accumulation of total phenols and flavonoids in the ‘Xiaobai’ apricot fruit.

Apricot fruit is susceptible to ripen progress during postharvest storage by the increasing of ROS production, which is an important reason of causing membrane damage, and the commercial quality of fruit would be reduced [12]. H_2_O_2_ and O_2_^.−^ are the most abundant ROS in fruits and vegetables, while MDA is the product of membrane lipid peroxidation that can reflect the membrane lipid oxidation and loss of permeability. The present study revealed that UV-C irradiation stimulated the accumulation of H_2_O_2_ and O_2_^.−^ in the early stages of storage and increased the degree of membrane lipid peroxidation in both UV-C and combined treatments. This could be due to UV-C treatment improving the respiratory metabolism of apricot fruit, which is one of the important sources of ROS accumulation [46]. It was previously reported that mitochondria produced a large amount of ROS when plants were exposed to UV-C irradiation [47]. Furthermore, Vandenabeele et al. [48] demonstrated that excessive ROS would damage cells, whereas a low concentration of ROS could act as a chemical signal. For example, ROS positively affects the accumulation of phenolics as a signaling role in fruits and vegetables [13,49]. Whether ROS will act as damaging or signaling molecule depends on the delicate equilibrium between ROS production and scavenging. In low concentrations, ROS act as signaling molecules that mediate several plant responses in plant cells, including responses under stresses [50]. For instance, ROS are produced in plants in response to drought stress, which would trigger oxidative stress and induce the ROS scavenging system that may confer protection or tolerance against stress that is emerging [51]. Therefore, maintaining the homeostasis of ROS in cells is very important for delaying the senescence of fruits after harvest. SOD can catalyze the O_2_^.−^ to H_2_O_2_, and CAT, POD, and APX can convert H_2_O_2_ to H_2_O [52]. Our findings demonstrated that UV-C treatment could effectively stimulate the activity of POD, CAT, SOD, and APX, as well as the expression of relative genes in apricot fruit at the initial stage of storage, and that the effect was superior to 1-MCP treatment, which could be attributed to the toxic excitatory effect of UV-C [53]. Limited accumulation of ROS due to UVC leads to the occurrence of oxidative stress in fruits, which stimulates the expression of antioxidant-related genes. Maurer et al. [13] revealed that UV-C treatment could increase the accumulation of ROS, which could stimulate the enzymatic antioxidant system in ‘Isabel’ grapes. Studies in strawberries suggested that UV-C irradiation could boost the activity of CAT, POD, and SOD [28]. Moreover, Xu et al. [27] reported that 1-MCP treatment efficiently maintained SOD and APX activity in kiwi fruit. The combined treatment appeared to significantly improve antioxidant enzyme activity, possibly due to the superimposed effect of 1-MCP and UV-C. Meanwhile, the occurrence of oxidative stress reaction gradually increased the antioxidant capacity of UV-C treated to fruit, resulting in the inhibition of MDA and H_2_O_2_ accumulation. Moreover, in the combined treatment group, ROS production was further inhibited by 1-MCP and UV-C, the accumulation of H_2_O_2_ and O_2_^.−^ was the lowest at the end of storage, and DPPH scavenging ability was stronger than the single treatment groups. Previous studies indicated that UV-C could boost the antioxidant capacity and inhibit ROS accumulation in mandarin [44]. Furthermore, Huan et al. [54] demonstrated that 1-MCP treatment inhibited the accumulation of H_2_O_2_ in peaches, and Ma et al. [55] identified that 1-MCP treatment improved the DPPH scavenging ability in ‘Jonagold’ apple. Thus, combining UV-C with 1-MCP treatment effectively improves antioxidant ability and reduces ROS damage in ‘Xiaobai’ apricot fruit.

In conclusion, 1-MCP treatment could inhibit the ethylene responses, reducing respiratory metabolism and restraining ROS accumulation, thereby delaying apricot fruit ripening and senescence. UV-C treatment, as abiotic stress, can potentially damage cells (particularly membrane and DNA) while promoting respiration metabolism. High levels of respiration metabolism provide the energy for cell repair and accelerate ROS accumulation. ROS can signal the antioxidant system, improving the activity of antioxidant enzymes and the content of secondary metabolites. The improved antioxidant ability could scavenge ROS and reduce oxidative damage, thereby delaying apricot fruit senescence (Figure 6). The combined treatment has the advantages of two treatments that can inhibit endogenous ethylene ripening and improve antioxidant capacity in ‘Xiaobai’ apricot fruit.

## 5. Conclusions

The results of our study reveal that application of UV-C irradiation combined with 1-MCP treatment was an effective measure to delay ripening and improve postharvest quality of ‘Xiaobai’ apricot fruit. The combined treatment could effectively delay the increasing of respiration rate, ethylene production, weight loss, and SSC; inhibit the decline of firmness, TA, and ASA; promote the accumulation of TP and flavonoids; improve the activity and expression of antioxidant related enzymes (POD, SOD, CAT, and APX) and DPPH scavenging ability; and reduce the production of MDA, H_2_O_2_, and O_2_^.−^. In conclusion, UV-C combined with 1-MCP treatment could be a viable strategy to prolong the storage period and improve the edible quality in postharvest apricot fruit. Moreover, the future research would focus on exploring the effect of UV-C irradiation combined with other preservation technologies, such as near-freezing temperature storage or coating preservation, in improving the quality of fruit and vegetables.

## Figures and Tables

**Figure 1 foods-12-01296-f001:**
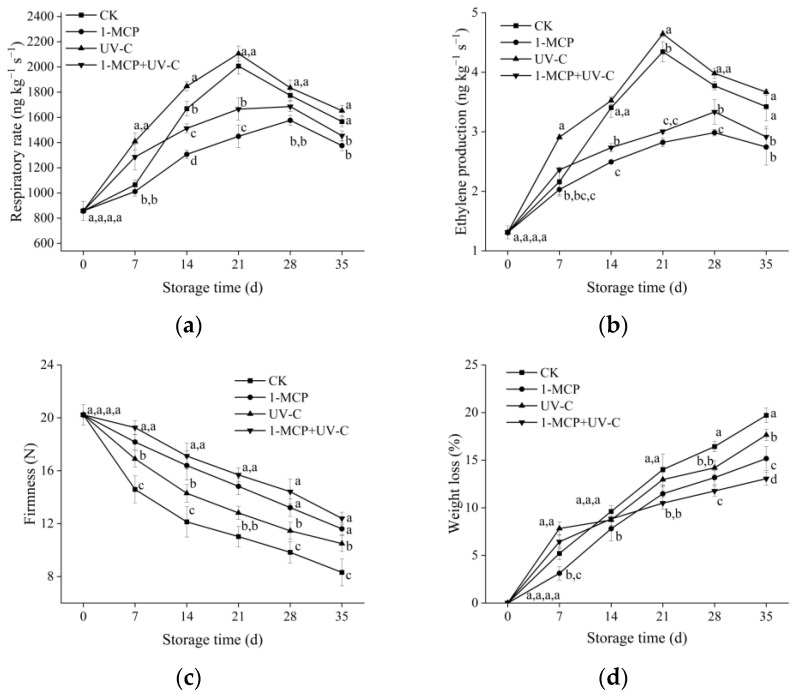
Effects of UV-C, 1-MCP, and their combination on respiratory rate (**a**), ethylene production (**b**), firmness (**c**), weight loss (**d**), soluble solid content, (**e**) and titratable acid (**f**) of apricot fruit during storage at 4 ± 1 °C. Each value is a mean of three replications ± SD. The error bars represent SD. The letters a, b, c, and d indicate statistically significant differences (*p* < 0.05) between the sample groups according to a Duncan’s multiple range test.

**Figure 2 foods-12-01296-f002:**
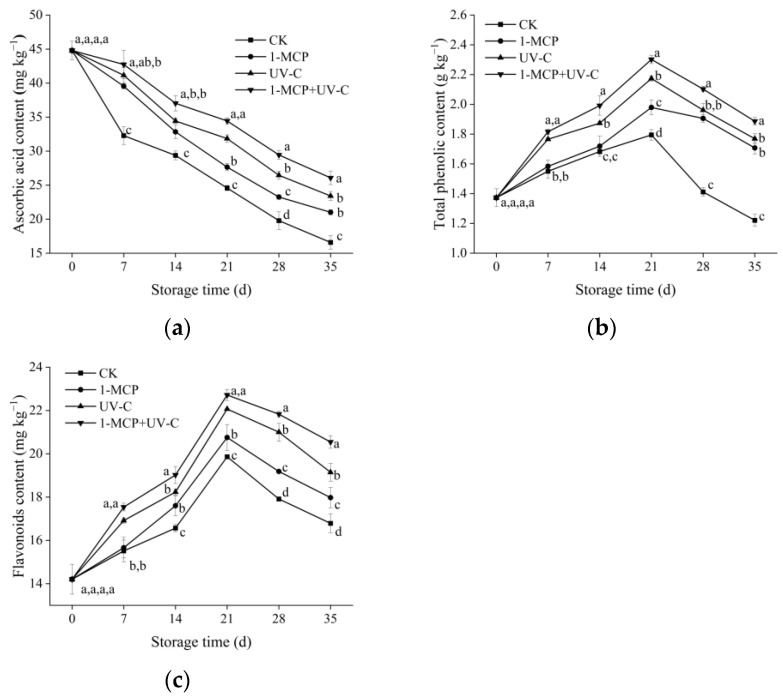
Effects of UV-C, 1-MCP, and their combination on ascorbic acid (**a**), total phenolics content (**b**), and flavonoid content (**c**) of apricot fruit during storage at 4 ± 1 °C. Each value is a mean of three replications ± SD. The error bars represent SD. The letters a, b, c, and d indicate statistically significant differences (*p* < 0.05) between the sample groups according to a Duncan’s multiple range test.

**Figure 3 foods-12-01296-f003:**
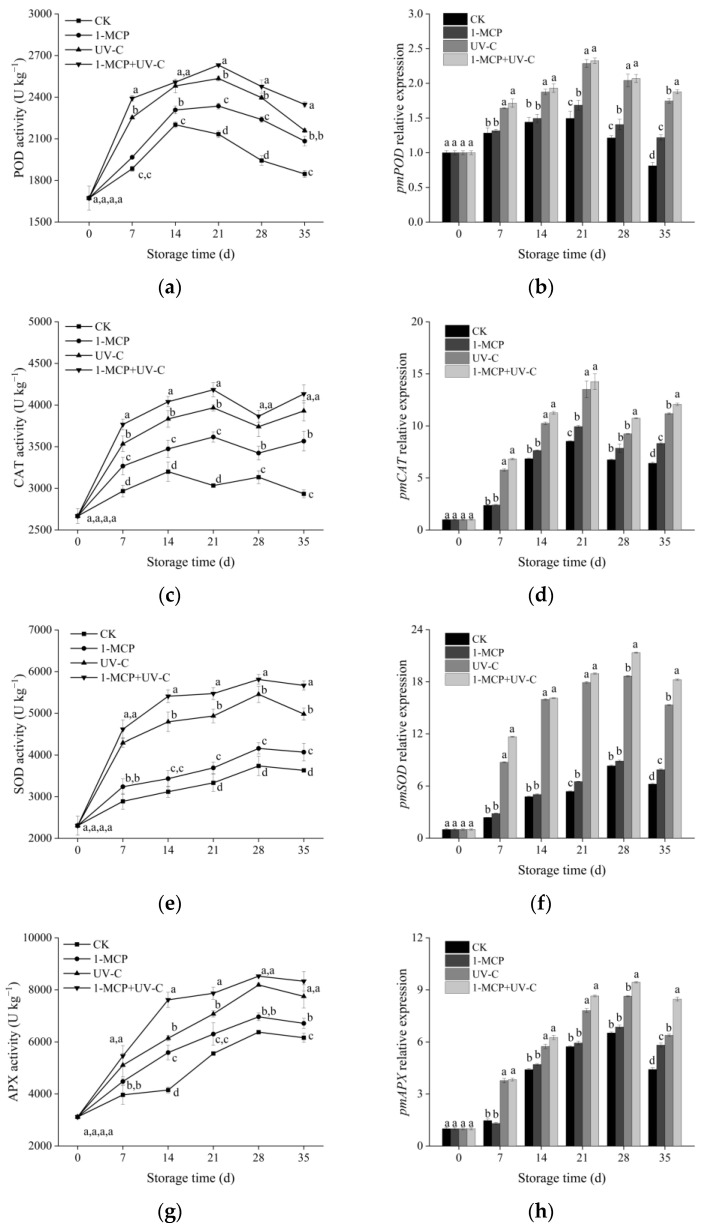
Effects of UV-C, 1-MCP, and their combination on peroxidase (POD) activity (**a**), *pmPOD* expression (**b**), catalase (CAT) activity (**c**), *pmCAT* expression (**d**), superoxide dismutase (SOD) activity (**e**), *pmSOD* expression (**f**), ascorbate peroxidase (APX) activity (**g**), and *pmAPX* expression (**h**) of apricot fruit during storage at 4 ± 1 °C. Each value is a mean of three replications ± SD. The error bars represent SD. The letters a, b, c, and d indicate statistically significant differences (*p* < 0.05) between the sample groups according to a Duncan’s multiple range test.

**Figure 4 foods-12-01296-f004:**
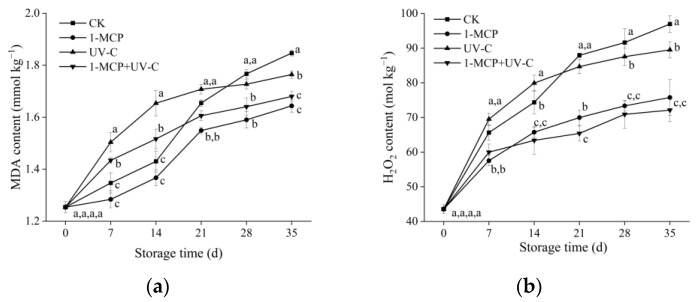
Effects of UV-C, 1-MCP, and their combination on MDA content (**a**), H_2_O_2_ content (**b**), O_2_^.−^ production rate, (**c**) and DPPH scavenging rate (**d**) of apricot fruit during storage at 4 ± 1 °C. Each value is a mean of three replications ± SD. The error bars represent SD. The letters a, b, c, and d indicate statistically significant differences (*p* < 0.05) between the sample groups according to a Duncan’s multiple range test.

**Figure 5 foods-12-01296-f005:**
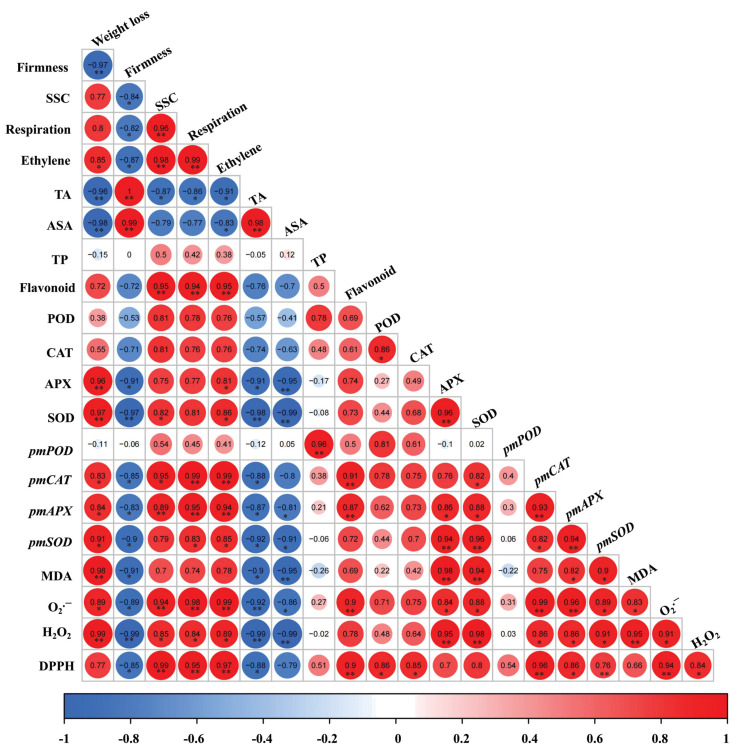
Pearson’s correlation matrix among the measured parameters in ‘Xiaobai’ apricot. Positive correlations are displayed in red and negative correlations in blue. Asterisks indicate a significant correlation between two indexes (* *p* < 0.05; ** *p* < 0.01).

**Figure 6 foods-12-01296-f006:**
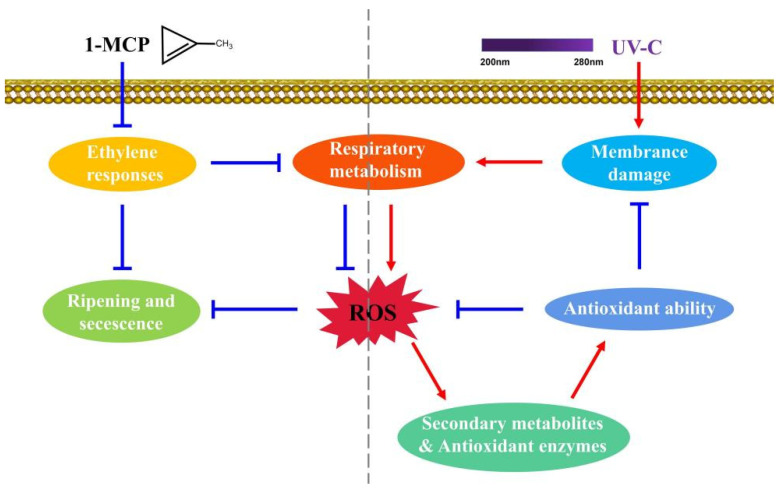
Diagrammatic model of the potential mechanism of UV-C and 1-MCP treatment on physicochemical characterization and antioxidant activity in ‘Xiaobai’ apricot. Red represents a positive effect, while blue represents a negative effect.

**Table 1 foods-12-01296-t001:** Primers of real-time quantitative PCR.

Gene Name	GeneBank ID	Forward Primers (5′-3′)	Reverse Primers (5′-3′)
*pmPOD*	XM_008238416.1	GCCAAGTTCACATCAAAGGGTC	GATAGGCGTCTCAAGTTTGTAA
*pmCAT*	XM_016795622.1	GATACTCAGAGGCACCGTCTTG	CACACTTCTCACGCTTTCCATAA
*pmSOD*	XM_008240505.1	ACCTGGGAACATTGGATACTGA	GCTTTGATACCAGAGTCCGACT
*pmAPX*	XM_008237784.2	GAGCTGTAGTTACGGTAAGG	TGTGCCCTTCCCAGTGTATGAC

## Data Availability

The data supporting the findings of this study are available within the article.

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
