# Peer review of "1-Methylcyclopropene and UV-C Treatment Effect on Storage Quality and Antioxidant Activity of ‘Xiaobai’ Apricot Fruit"

_foods, 2023, doi:10.3390/foods12061296_

Round 1

Reviewer 1 Report

The manuscript entitled "1-Methylcyclopropene and UV-C treatment effect on storage quality and antioxidant activity of ‘Xiaobai’ apricot fruit" by Lv et al. provides supporting data regarding the combined effect of UV-C and 1-MCP upon the post-harvest physiology of apricot fruit.

The overall text is easy to read, well written and the presentation of the results facilitates the reader to understand the scope of the project and the overall effect of the treatments upon the fruit.

In the abstract, lines 10-11, the authors should indicate that this study was focused upon the combined effect of the application of 1-MCP and UV-C upon apricot fruit prior to their storage.

The Introduction is very well written.

In the Materials and Methods section, do indicate how the enzymatic activity is expressed. U g-1 or U Kg-1? cause in the text (lines 283-284) and Figure 3 (a), different values are presented.

The results section is very well written, and all the Figures are easy to read and well organized.

In the discussion section, line 403, it is not "effectively improve" but "effectively stimulate".

Also, in line 420, it is "loss of permeability". Furthermore, in line 423, "UV-C treatment accelerated the ".

Author Response

Thank you for your valuable advice. The errors has been corrected in text.

In the Materials and Methods section, do indicate how the enzymatic activity is expressed. U g-1 or U Kg-1? cause in the text (lines 283-284) and Figure 3 (a), different values are presented.

‘Expressed with unit of U kg-1’ was added in text.

In the discussion section, line 403, it is not "effectively improve" but "effectively stimulate".

The text has been revised.

Also, in line 420, it is "loss of permeability". Furthermore, in line 423, "UV-C treatment accelerated the ".

The text has been revised.

Reviewer 2 Report

Dear Authors,

The topic of the paper is not new, as similar manuscripts showing the effect of 1-methylcyclopropene and UV-C irradiation have been published previously (for example, Pristijono et al., 2018, J Food Sci Technol 55:7, 2467 or Xu & Liu 2017, Food Bioprocess Technol, 10, 1695, the latter you cited as [35]). The final conclusions in those papers were the same as in the manuscript proposed by the authors, the only difference being the object of study (apricot, not blueberry or lime). Hence, I surmised that the scientific soundness of the work was rather low, since I do not see the novelty of the work.

In addition, Authors published a paper on the effect of 1-meethylcyclopropene on the quality of the ‘Xiaobai’ apricot fruit in 2021 (2nd position in the reference list) which was cited in line 28 but should rather be cited in lines 52-54. The proposed article is very similar in the experimental design to previous papers, only an additional factor (UV-C) was introduce, but as I mentioned above this is not new.

Methods

Please put the date of the harvest.

The description regarding the statistical analysis is missing. Why was Duncan's test used for post-hoc analysis?

Results.

Lines 265-267. Please check the results in figure (3a) and in the text. There are significant differences between control and 1-MCM treated sample on day 14 (indices c and b, respectively).

Line 273, delete ‘however’ better is  to use ‘moreover’

Line 281: the decrease was not statistically significant for each sample

Lines 335-344

For some of the correlations shown, I have doubts that the relationship between the data was linear to calculate linear r-Pearson correlation coefficients. Besides, the significance of the correlations was not shown, and it is known that the sample size has a huge impact on the p-value in the correlation analysis. With such data, it is better to show regression analysis and significant differences in regression coefficients. In addition, some kind of multivariate analysis would show the relationships between variables. I am curious if it is possible to differentiate the sample with different treatments using the parameters shown (in the cluster analysis or discriminant analysis).

Discussion

Line 394: Definitely this is not a synergistic effect (it could be calculated from the data).

Lines 400-412: relations between enzymes expression and ROS production is quite interesting and should be extended to put more insight into the mechanism (to this end PCA analysis in the result section would be useful)

Line 429: how did you prove the synergistic effect?

Conclusions

I have doubt if UV-C treatment in combination with 1-MCM gives better results than 1-MCM alone (respiratory rate, TA, SSD, H2O2).

Lines 460-461: Is this statement true? For enzyme activity and TPC , TFC and AAC the combined treatment seems to be not different from single UV-C treated sample. See your text in line 392 and results section.

Line 465: What technologies? List them.

Author Response

Thank you for your valuable comments, the modification had been revised in text.

In addition, Authors published a paper on the effect of 1-meethylcyclopropene on the quality of the ‘Xiaobai’ apricot fruit in 2021 (2nd position in the reference list) which was cited in line 28 but should rather be cited in lines 52-54. The proposed article is very similar in the experimental design to previous papers, only an additional factor (UV-C) was introduce, but as I mentioned above this is not new.

Response: "line 28" is to introduce the adverse effects of special geographical and climatic conditions in Xinjiang on the storage of ‘Xiaobai’ apricot, which has certain specificity, while lines 52-54 mainly introduces the principle of 1-MCP, so it is necessary to quote a more authoritative article.

Methods

Please put the date of the harvest.

Response: The date of the harvest was Jun 2020, and was added in text.

The description regarding the statistical analysis is missing. Why was Duncan's test used for post-hoc analysis?

Response: Due to our negligence in checking the article, the statistical analysis was not added, which has been added.

Results.

Lines 265-267. Please check the results in figure (3a) and in the text. There are significant differences between control and 1-MCM treated sample on day 14 (indices c and b, respectively).

Response: The text has been revised.

Line 273, delete ‘however’ better is  to use ‘moreover’

Response: The text has been revised.

Response: Line 281: the decrease was not statistically significant for each sample

Response: The text has been revised.

Lines 335-344 For some of the correlations shown, I have doubts that the relationship between the data was linear to calculate linear r-Pearson correlation coefficients. Besides, the significance of the correlations was not shown, and it is known that the sample size has a huge impact on the p-value in the correlation analysis. With such data, it is better to show regression analysis and significant differences in regression coefficients. In addition, some kind of multivariate analysis would show the relationships between variables. I am curious if it is possible to differentiate the sample with different treatments using the parameters shown (in the cluster analysis or discriminant analysis).

Response: The p-value in the correlation analysis was added as ‘*’ and ‘**’ which indicate p< 0.05 and p< 0.01, respectively.

Discussion

Line 394: Definitely this is not a synergistic effect (it could be calculated from the data).

Response: The ‘synergistic effect’ has been modified to ‘due to that the secondary metabolism and antioxidant capacity of fruit was enhanced by UV-C.

Lines 400-412: relations between enzymes expression and ROS production is quite interesting and should be extended to put more insight into the mechanism (to this end PCA analysis in the result section would be useful)

Response: ROS in low concentrations acted as signaling molecules that mediate several plant responses in plant cells, including responses under stresses, which could induce the expression of antioxidant enzymes.

Line 429: how did you prove the synergistic effect?

Response: ‘The synergistic effect’ is not rigorous and had been deleted. The whole sentence had been modified.

Conclusions

I have doubt if UV-C treatment in combination with 1-MCM gives better results than 1-MCM alone (respiratory rate, TA, SSD, H2O2).

Response: Indeed, the combined treatment was inferior to 1-MCP treatment in respiration rate, but superior to 1-MCP treatment in secondary metabolite accumulation and antioxidant capacity.

Lines 460-461: Is this statement true? For enzyme activity and TPC , TFC and AAC the combined treatment seems to be not different from single UV-C treated sample. See your text in line 392 and results section.

Response: There are significant difference between combined treatment and UV-C treatment on day 28 and 35 in TPC, TFC and ASA.

Line 465: What technologies? List them.

Response: The text has been revised.

Round 2

Reviewer 2 Report

Dear Authors,

Statistical analysis still needs improvements. You did not answer my question: Why was the Duncan test used for post-hoc analysis and not the Tukey's test? There is no information about the method testing significant effects. There is lack of information on homogeneity of variance in the dataset.

Please answer my previous comment: Lines 335-344 For some of the correlations shown, I have doubts that the relationship between the data was linear to calculate linear r-Pearson correlation coefficients. Besides, the significance of the correlations was not shown, and it is known that the sample size has a huge impact on the p-value in the correlation analysis. With such data, it is better to show regression analysis and significant differences in regression coefficients. In addition, some kind of multivariate analysis would show the relationships between variables. I am curious if it is possible to differentiate the sample with different treatments using the parameters shown (in the cluster analysis or discriminant analysis).

Please improve the manuscript in line with my previous comment: Lines 400-412: relations between enzymes expression and ROS production is quite interesting and should be extended to put more insight into the mechanism; You did not extend that section just change the order of paragraphs and sentences.

Please improve conclusions (see my previous comments)

Author Response

Dear Reviewer,

Thank you for your valuable advice. The answers to the comment are as follows:

Statistical analysis still needs improvements. You did not answer my question: Why was the Duncan test used for post-hoc analysis and not the Tukey's test? There is no information about the method testing significant effects. There is lack of information on homogeneity of variance in the dataset.

The least significant difference (LSD) and Duncan test are two major method used for post-hoc analysis to analyze significant differences, as referred to https://doi.org/10.1016/j.foodchem.2019.125726; https://doi.org/10.1111/jfbc.12857 and https://doi.org/10.1016/j.foodchem.2019.03.024; ttps://doi.org/10.1016/j.postharvbio.2021.111613; https://doi.org/10.1016/j.foodchem.2020.127981, and I choose Duncan test as the method of analyzing significant differences.

And the ‘2.9. Statistical analysis’ was added in lines 204-209.

Please answer my previous comment: Lines 335-344 For some of the correlations shown, I have doubts that the relationship between the data was linear to calculate linear r-Pearson correlation coefficients. Besides, the significance of the correlations was not shown, and it is known that the sample size has a huge impact on the p-value in the correlation analysis. With such data, it is better to show regression analysis and significant differences in regression coefficients. In addition, some kind of multivariate analysis would show the relationships between variables. I am curious if it is possible to differentiate the sample with different treatments using the parameters shown (in the cluster analysis or discriminant analysis).

My apologies, the correlation analysis was calculated by Pearson correlation, and the relevant explanation had been added in fig 5. And the p-value was added in fig 5 which was expreed as Asterisks (*p < 0.05; **p<0.01).

Please improve the manuscript in line with my previous comment: Lines 400-412: relations between enzymes expression and ROS production is quite interesting and should be extended to put more insight into the mechanism; You did not extend that section just change the order of paragraphs and sentences.

Further changes have been made to the discussion section, and additions and modifications have been highlighted.

Please improve conclusions (see my previous comments)

The sentence of ‘The effect of combined treatment in delaying ripening and improving postharvest quality was more advantageous than that of UV-C or 1-MCP treatment alone’ had been deleted.